# Pixel Super-Resolution Phase Retrieval for Lensless On-Chip Microscopy via Accelerated Wirtinger Flow

**DOI:** 10.3390/cells11131999

**Published:** 2022-06-22

**Authors:** Yunhui Gao, Feng Yang, Liangcai Cao

**Affiliations:** State Key Laboratory of Precision Measurement Technology and Instruments, Department of Precision Instruments, Tsinghua University, Beijing 100084, China; gyh21@mails.tsinghua.edu.cn (Y.G.); yangfeng2020@mail.tsinghua.edu.cn (F.Y.)

**Keywords:** phase retrieval, pixel super-resolution, computational imaging, digital holography, Wirtinger flow, lensless imaging, whole slide imaging

## Abstract

Empowered by pixel super-resolution (PSR) and phase retrieval techniques, lensless on-chip microscopy opens up new possibilities for high-throughput biomedical imaging. However, the current PSR phase retrieval approaches are time consuming in terms of both the measurement and reconstruction procedures. In this work, we present a novel computational framework for PSR phase retrieval to address these concerns. Specifically, a sparsity-promoting regularizer is introduced to enhance the well posedness of the nonconvex problem under limited measurements, and Nesterov’s momentum is used to accelerate the iterations. The resulting algorithm, termed accelerated Wirtinger flow (AWF), achieves at least an order of magnitude faster rate of convergence and allows a twofold reduction in the measurement number while maintaining competitive reconstruction quality. Furthermore, we provide general guidance for step size selection based on theoretical analyses, facilitating simple implementation without the need for complicated parameter tuning. The proposed AWF algorithm is compatible with most of the existing lensless on-chip microscopes and could help achieve label-free rapid whole slide imaging of dynamic biological activities at subpixel resolution.

## 1. Introduction

The ever-increasing demand for information throughput toward biomedical and other engineering applications has strongly promoted recent developments in imaging techniques with a high space-bandwidth product [1,2,3]. Among these techniques, lensless on-chip microscopy has become an emerging solution by leveraging recent advances in sensor technology and computational power to overcome the inherent tradeoff between spatial resolution and field of view of the conventional point to-point imaging modality [4,5,6]. It captures a field-of-view as large as the sensor area with a resolution of a few microns in a single exposure, which makes it a competitive solution to rapid whole slide imaging for histopathology [7]. Compared with the lens-based counterparts, lensless on-chip microscopy also enables a compact and low-cost configuration [8,9]. Furthermore, its quantitative phase imaging capability allows for the label-free characterization of transparent or volumetric samples [10], which are commonly encountered in biomedical applications, such as pathology [11], inflammation [12], immunology [13], neuroscience [14], and cancer cell biology [15].

Despite their distinct advantages, lensless on-chip microscopy poses new challenges that need to be addressed. The phase information of the wavefield cannot be recorded due to the intensity-only response of the imaging sensors, and the high-frequency details beyond the Nyquist sampling limit are also lost during the measurement. Techniques to address these problems, namely phase retrieval [16] and pixel super-resolution (PSR) [17], play a pivotal role in high-fidelity and high-resolution holographic imaging. Phase retrieval aims to encode the phase of the wavefield by transferring it into intensity variations by physical means [18,19,20,21] and then numerically recover the phase (and amplitude) distributions via optimization algorithms [22,23,24]. Pixel super-resolution, on the other hand, aims to surpass the Nyquist sampling limit by similar physical encoding and numerical recovery procedures, pushing the resolution toward the diffraction limit [25,26,27,28,29,30].

In recent years, it has been recognized from a physical perspective that the missing phase and the undersampled high-frequency information can be both encoded into the intensity observations via diversity measurements [31,32,33], which can be implemented by varying the defocus distances [34,35,36], illumination wavelengths [37,38,39,40], modulation patterns [41,42,43], and probe positions [44,45,46], etc., making it possible for numerical recovery. On the algorithmic side, PSR phase retrieval can be achieved by a simple modification to the classical phase retrieval algorithms [47,48,49]. More recently, PSR phase retrieval has been recast as a standard optimization problem, which allows the use of off-the-shelf optimization tools, such as alternating projection and gradient descent algorithms [33].

Nevertheless, the range of applications of current PSR phase retrieval methods is primarily limited by the considerable time consumption during both the measurement and reconstruction stages. Due to the high dimension of the parameter space, it typically requires a larger number of diversity measurements to ensure the well posedness of the inverse problem and takes a longer time for iterative reconstruction compared with classical phase retrieval methods, yet acquisition and reconstruction speed is of vital importance in many imaging applications [50,51]. Additionally, the processing of the data is further complicated by manual parameter tuning due to the heuristic nature of the algorithms.

In this work, we introduce accelerated Wirtinger flow (AWF) as a unified framework for pixel super-resolution phase retrieval. Based on the proximal gradient method, AWF allows incorporating off-the-shelf regularization techniques to help improve imaging quality and reduce measurement number. Nesterov’s acceleration method is applied in the iterative reconstruction process, achieving at least an order of magnitude faster rate of convergence. Furthermore, the proposed algorithm features a prespecified step size, facilitating simple implementation to various system configurations.

## 2. Problem Formulation

### 2.1. Forward Model

Figure 1a shows some typical optical setups for a lensless on-chip microscope. The sample is illuminated by a coherent source. Diversity measurement is achieved by varying the physical parameters. The intensity of the coded wavefield is then recorded by the pixelated sensor. Therefore, the general forward model for lensless on-chip microscopes can be expressed as a linear transformation and a down-sampled quadratic measurement:(1)$$y_{k}^{2}=S{\left|A_{k}x\right|}^{2},\;k=1,2,\cdots ,K,$$
where $x\in C^{n}$ represents the complex transmittance of the sample, $A_{k}\in C^{m\times n}$ denotes the sampling matrix for the *k*-th of out *K* diversity measurements, and $y_{k}^{2}\in R^{d}$ denotes the corresponding intensity image. $S\in R^{d\times m}$ with m=σd represents the down-sampling (pixel binning) operation of the sensor pixels, where σ is a positive integer referred to as the down-sampling ratio. The down-sampling operator performs a weighted sum of the subpixel intensities. Conceptual illustrations of the physical model and the mathematical model are shown in Figure 1 and Figure 2, respectively. Note that ${\left(\cdot \right)}^{2}$ and |·| are element-wise operators.

While the forward measurement matrix $A_{k}$ given in Equation (Equation 1) seems rather abstract, it in fact encapsulates a wide variety of physical processes. For example, when using defocus diversity or wavelength diversity measurements, $A_{k}$ may represent the free-space propagation with multiple distances or wavelengths. In the case of ptychography or modulation diversity, it may also incorporate coded illumination or mask modulation.

The introduction of the down-sampling matrix S makes the forward model of Equation (Equation 1) different from that of classical phase retrieval problems. It arises from the fact that all the photons incident upon a same pixel are converted into a single intensity signal. Mathematically, the intensity signal can be regarded as a weighted sum of the signals of the corresponding subpixels. In many works, uniform weights are adopted. Nevertheless, in practice, the fill factor of the sensor pixels is usually less than one, thus the weights for different subpixels may be different. A more accurate sampling model can be obtained by experimentally calibrating the intensity response of the sensor pixels, as was well done in [7]. Note that this down-sampling process is quite similar to the single pixel imaging model, despite the fact that we are using array sensors [52]. The model described by Equation (Equation 1) subsumes a special case the classical phase retrieval problem when σ=1 and S is an identity matrix.

### 2.2. Regularized Inversion

Based on the forward model of Equation (Equation 1), PSR phase retrieval can be formulated as a regularized inverse problem as follows:(2)x^=argminx11112K∑k=1KSAkx2−yk22⏟F(x)+111λDx1⏟R(x),
where F(x) and R(x) are the data-fidelity function and the regularization function, respectively. ${∥\cdot ∥}_{p}$ denotes the $\ell_{p}$ vector norm. The data-fidelity function ensures the estimate x is consistent with the forward model. Considering the inherent ill posedness of PSR phase retrieval, an additional regularization term R(x) is introduced that encourages certain solutions based on prior knowledge of the sample. We use the anisotropic complex total variation (TV) in this work as an example, where $D\in R^{2n\times n}$ denotes the finite difference operator and λ>0 is a regularization parameter. When λ=0, Equation (Equation 2) reduces to the non-regularized case.

It should be noted that there exists various mathematically equivalent choices for the data-fidelity function. The same issue has been well studied in the case of classical phase retrieval. The *intensity-based* formulation aims to minimize the intensity residuals, which was adopted by many theoretical studies [24,53,54]. Another choice is the *amplitude-based* formulation [55,56,57,58]. It has been observed by many prior works that minimizing the lower-order amplitude-based fidelity function leads to faster convergence compared with minimizing the intensity-based one [59]. In analogy to the classical phase retrieval, the lower-order fidelity function in Equation (Equation 2) was adopted in [33] for PSR phase retrieval, whose superior performance was also experimentally verified.

The regularization term R(x) encourages certain solutions based on prior knowledge of the sample. In classical phase retrieval, chances of recovering the unknown sample rely heavily on the well posedness of the problem. Broadly speaking, there are two ways to ensure well posedness, that is, increasing the number of measurements, or incorporating signal priors [60]. In the case of PSR phase retrieval, it is generally more time consuming to acquire sufficient diversity data to suppress ambiguous solutions. As a result, to tackle the ill-posedness, we introduce another regularization term R(x) for the reconstruction problem. Apart from the total variation function adopted in this paper, the regularization function can take many other forms, such as BM3D [42,61,62,63] and deep denoiser priors [64,65].

## 3. Derivation of Algorithms

### 3.1. Accelerated Wirtinger Flow

The proximal gradient method is adopted for solving the non-smooth composite optimization problem of Equation (Equation 2), which proceeds by minimizing the two terms in an alternative manner [66]. Specifically, we apply a gradient update step with respect to the fidelity term, whose Wirtinger gradient is given by [33]
(3)$$\nabla_{x}F\left(x\right)=\frac{1}{2K}\sum_{k=1}^{K}A_{k}^{H}\mathrm{diag}\left(A_{k}x\right)S^{T}\left(1-\frac{y_{k}}{\sqrt{S{\left|A_{k}x\right|}^{2}}}\right),$$
where ${\left(\cdot \right)}^{T}$ and ${\left(\cdot \right)}^{H}$ denote the transpose and Hermitian operators, respectively. diag(·) puts the element of a vector onto the diagonal of a matrix. $\sqrt{\cdot }$ and ·/· are element-wise operators. Rigorously speaking, the fidelity function is not differentiable at points where $S|A_{k}{x|}^{2}$ has zero entries for some *k*. However, its non-smoothness can be addressed by assigning a certain value to the gradients at these points. The regularization term is updated via its proximity operator:(4)$${\mathrm{prox}}_{\gamma R}\left(v\right)=\underset{x}{\mathrm{argmin}}\frac{1}{2\gamma }{∥x-v∥}_{2}^{2}+R\left(x\right),$$
where γ>0 denotes the step size. For the complex TV function, in particular, an efficient algorithm for calculating the proximal update has been proposed in [67]. We found that when the regularization parameter λ is relatively small (as is often the case), a single iteration of the inner loop is sufficient for good performance.

The basic proximal gradient algorithm, however, can be very slow in terms of convergence. Fortunately, relating the existing PSR phase retrieval algorithms with theoretically tractable optimization frameworks allows us to explore advanced techniques that help improve the algorithmic performance. In this work, we introduce the well-known Nesterov’s acceleration method as an example. It was originally proposed for minimization of smooth convex functions [68], and was later extended to the proximal gradient method [69]. Inspired by Nesterov’s method, some recent works have applied similar acceleration schemes to classical phase retrieval [70,71,72,73]. Following this line of research, we introduce a similar acceleration scheme to PSR phase retrieval, leading to the following iterates: (5)$$\begin{array}{cc} v^{\left(t\right)}= & \;u^{\left(t-1\right)}-\gamma \nabla_{u}F\left(u^{\left(t-1\right)}\right), \end{array}$$(6)$$\begin{array}{cc} x^{\left(t\right)}= & \;{\mathrm{prox}}_{\gamma R}\left(v^{\left(t\right)}\right), \end{array}$$(7)$$\begin{array}{cc} u^{\left(t\right)}= & \;x^{\left(t\right)}+\beta_{t}\left(x^{\left(t\right)}-x^{\left(t-1\right)}\right), \end{array}$$
where t=1,2,⋯,T, γ>0 is the step size, and $u^{\left(0\right)}=x^{\left(0\right)}$. The algorithm is termed as accelerated Wirtinger flow when $\beta_{t}=t/\left(t+3\right)$ is used for the the extrapolation step of Equation (Equation 7), as is suggested by Nesterov’s method.

### 3.2. Convergence Analysis

We next provide general guidance for the step size selection of the AWF algorithm. Considering the nonconvexity of the PSR phase retrieval problem, we present a weaker theoretical result establishing the convergence of the non-accelerated Wirtinger flow algorithm, which is summarized by the following theorem. Nevertheless, we empirically observe a stable convergence behavior of the AWF algorithm using the same step size. A detailed proof of the theorem can be found in the Appendix A document.

**Convergence** **Theorem** **1.**
*The Wirtinger flow iterates of Equations (Equation 5)–(Equation 7) with $\beta_{t}\equiv 0$ converge to a stationary point using a fixed step size*

γ

*that satisfies*

(8)
$$\left.\gamma \leq 2K/\sum_{k=1}^{K}\rho \left(A_{k}^{H}\mathrm{diag}\left(s\right)A_{k}\right),\right.$$

*where*

$s=S^{T}\cdot 1$

*, and*

ρ(·)

*denotes the spectral radius.*


## 4. Experimental Results

### 4.1. System Configuration

To validate the proposed AWF method, we consider a particular holographic imaging model based on phase modulation diversity, as shown in Figure 3. A phase-only spatial light modulator (SLM) is placed at the conjugate plane of a 4f system with respect to the sample, generating phase diversity by varying the modulation patterns. The diffraction patterns of the modulated wavefield are recorded by a CMOS sensor, which is placed close to the sample. Using SLM for diversity measurement enables higher data acquisition speed compared with approaches that require mechanical displacements. Phase-only liquid crystal SLMs can typically achieve a frame rate of 60 Hz [74]. Thus, measurements can be completed in less than a few seconds.

Based on the optical setup, we now formulate the specific mathematical model and derive the step size for the algorithm. The sampling matrix $A_{k}$ can be divided into three linear operations, namely a phase-only modulation by the SLM $M_{k}\in C^{n\times n}\;\left(k=1,2,\cdots ,K\right)$, a free-space propagation $H\in C^{n\times n}$ which is implemented via circular convolution based on the angular spectrum method, and an image cropping operation due to the finite size of the sensor area $C\in R^{m\times n}$. That is, we have $A_{k}={\mathrm{CHM}}_{k}$. As for the down-sampling operator S, we assume a spatially uniform weight for different subpixel responses. A more accurate sampling model can be obtained by experimentally calibrating the intensity response of the sensor pixels. Based on the above modeling, one can easily verify that $\rho (A_{k}^{H}\mathrm{diag}\left(s\right)A_{k})\leq 1$ for all k=1,2,⋯,K (see Appendix A document), which, according to the above theorem, implies a proper step size of γ=2. It is worth noticing that this specific choice of step size is nontrivial and generally applicable. For most optical settings, the measurement is passive in the sense that the sampling operators $A_{k}$ are non-expansive (i.e., $\rho (A_{k})\leq 1$) after proper normalization. Multiplication by a factor of two is due to the Wirtinger calculus [75].

To facilitate further applications and comparisons, a MATLAB implementation for the algorithms is available in [76]. All numerical experiments were conducted on a laptop computer equipped with an Intel Core i5 CPU at 1.60 GHz and 16 GB of memory. It takes approximately 20 s per iteration for reconstruction of a pixel super-resolved image of size n=1024×1024 using K=64 diversity images.

### 4.2. Simulation Studies

Numerical studies were conducted to quantitatively study the performance improvements in terms of both reconstruction quality and convergence speed. The *Cameraman* image and the *Peppers* image were used to simulate the amplitude and phase distribution of a complex sample, respectively. Only K=8 intensity images with phase modulation diversity were used for PSR phase retrieval with an under-sampling ratio of σ=4×4=16, rendering the inverse problem severely ill-posed. Figure 4a shows the retrieved amplitude and phase via the AWF algorithm with and without the TV regularization term. The introduction of the regularizer helps significantly suppress the artifacts while preserving fine details of the image. Furthermore, the TV-regularized reconstruction with K=8 images also outperforms the non-regularized reconstruction with K=32 images, as is quantified by the root-mean-square errors (RMSEs). Figure 4b plots the convergence curves of the AWF algorithm and the non-accelerated proximal gradient algorithm. We observe that in both non-regularized and regularized cases, AWF exhibits at least an order of magnitude faster rate of convergence compared with the basic algorithm, which empirically demonstrates the effectiveness of Nesterov’s method for this nonconvex optimization problem.

### 4.3. Optical Experiments

Experimental data were collected from an inline holographic imaging system, where a 532 nm laser was used for coherent illumination, a phase-only reflective SLM (GAEA-2, HOLOEYE, Berlin, Germany) was used to generate phase patterns, and a CMOS sensor (QHY163M, pixel pitch 3.8 μm, QHYCCD, Beijing, China) was used to record the intensity images. Readers may refer to [41] for a detailed description of the system configuration. The phase response of the SLM was calibrated by a self-referenced interferometric method [77,78,79]. The phase modulation patterns were randomly generated and then Gaussian-filtered in order to introduce enough diversity while minimizing the crosstalk effect between adjacent SLM pixels.

We first evaluated the proposed method via imaging a quantitative phase microscopy target (Benchmark Technologies, Lynnfield, MA, USA). The up-sampling ratio was set to σ=3×3, leading to a higher imaging resolution than the classical non-PSR phase retrieval method. Reconstruction via the non-regularized and the TV-regularized PSR phase retrieval models was implemented using K=8, 16, 32, and 64 intensity images, respectively. In the optical experiments, the artifacts arise not only from the measurement noise, but also from the modeling errors of the imaging system that are inevitable in practice. The TV regularization can help reduce the artifacts while preserving high-frequency details, as is visually demonstrated by the phase images and quantitatively verified by the cross-sectional phase profiles in Figure 5a. The ground truth values of the phase structures were calculated by φ=2πh(RI−1)/w, where *h* denotes the height of the structure, RI denotes the refractive index of the medium, and *w* denotes the illumination wavelength. Figure 5b indicates a similar improvement to the convergence rate using AWF compared with WF. Considering the inevitable modeling errors of the imaging system, the algorithms tend to converge earlier on experimental data.

We further tested the resolving power of the proposed method on the imaging of a biological sample under the same optical settings. Figure 6 shows the imaging results of an iron-hematoxylin stained slide of the uterus of *parascaris equorum*. The piece-wise smoothness property of the TV regularizer helps suppress the noise and artifacts significantly, although the reconstruction quality may be further improved by more advanced image priors. The shape of the chromosomes can be resolved by the PSR phase retrieval method with high fidelity, from which one could easily recognize different phases of the mitosis. In prephase (Figure 6b), the chromatin condenses into chromosomes. In metaphase (Figure 6c), the chromosomes line up along the equatorial plane. In anaphase (Figure 6d), two sets of daughter chromosomes are pulled toward opposite ends of the cell. Finally, in telophase (Figure 6e), a new envelope forms around each set of separated daughter chromosomes, and cell division occurs. These biological activities, however, are not clearly revealed by the conventional non-PSR phase retrieval method because the distribution of the chromosomes cannot be directly resolved by the relatively large sensor pixels.

## 5. Conclusions

To conclude, we proposed AWF as a general pixel super-resolution phase retrieval framework for lensless on-chip microscopy that helps reduce time consumption during both the measurement and reconstruction procedures. To speed up the data acquisition, we introduced the TV regularizer to tackle the ill posedness of PSR phase retrieval. As is demonstrated by both simulated and experimental data, the introduction of the TV regularizer allows at least a twofold reduction in the number of intensity images while maintaining competitive resolution and quality. To speed up the iterative reconstruction, we applied Nesterov’s method to the nonconvex PSR phase retrieval problem. The accelerated algorithm converges at least an order of magnitude faster than the conventional one. On the theoretical side, we demonstrated through theoretical analyses that the lower-order fidelity function has favorable geometrical properties that ensure convergence of the Wirtinger flow iterates using a prespecified step size. Our findings extend previous results on classical phase retrieval to the PSR case, which may help bridge the gap between empirical and theoretical studies. The proposed algorithmic framework is generally applicable to the existing lensless on-chip microscopy platforms, and may thus facilitate a wide range of biomedical applications, such as whole slide histopathology and cell biology.

## Figures and Tables

**Figure 1 cells-11-01999-f001:**
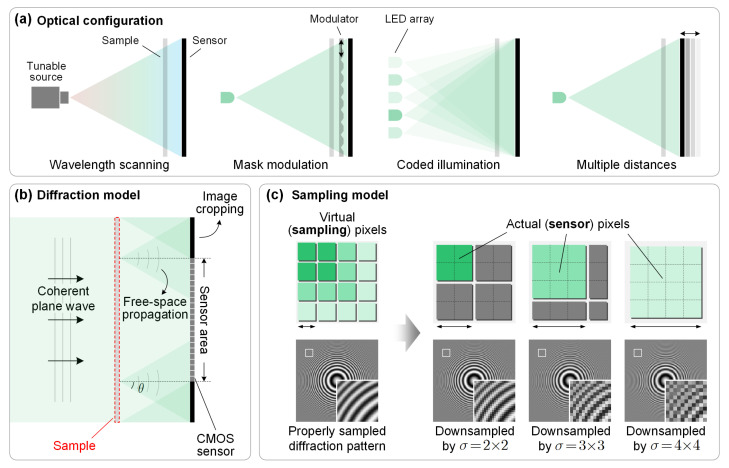
Forward model of a lensless on-chip microscope. (**a**) Typical optical configurations that can transfer the phase and subpixel information into the intensity variations at the sensor plane. (**b**) Diffraction model of the imaging system. Diffraction is calculated via the angular spectrum method, where the diffraction angle θ and the corresponding Fresnel kernel size are determined by the sampling frequency. (**c**) Sampling model of the sensor pixels.

**Figure 2 cells-11-01999-f002:**
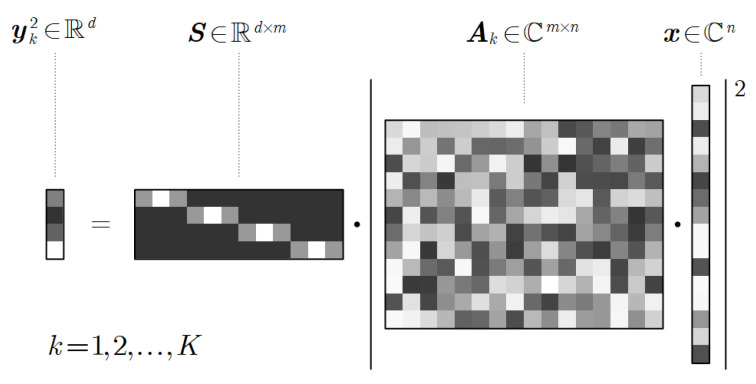
Intuitive illustration of the mathematical model.

**Figure 3 cells-11-01999-f003:**
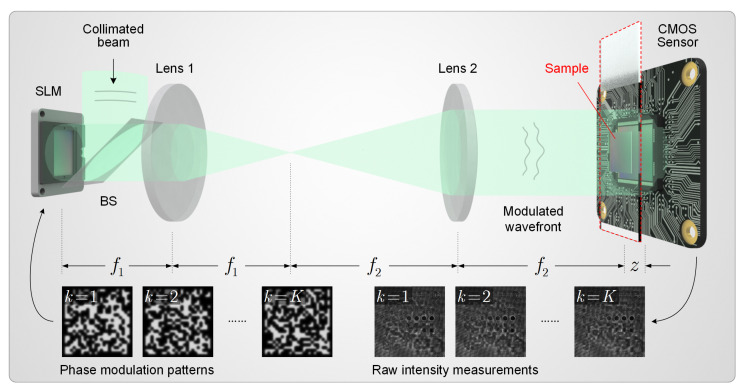
Lensless on-chip microscope based on phase modulation diversity, which we consider as an example in this work. BS is a beam splitter. $f_{1}$ and $f_{2}$ denote the focal lengths of Lens 1 and Lens 2, respectively. *z* denotes the sample-to-sensor distance.

**Figure 4 cells-11-01999-f004:**
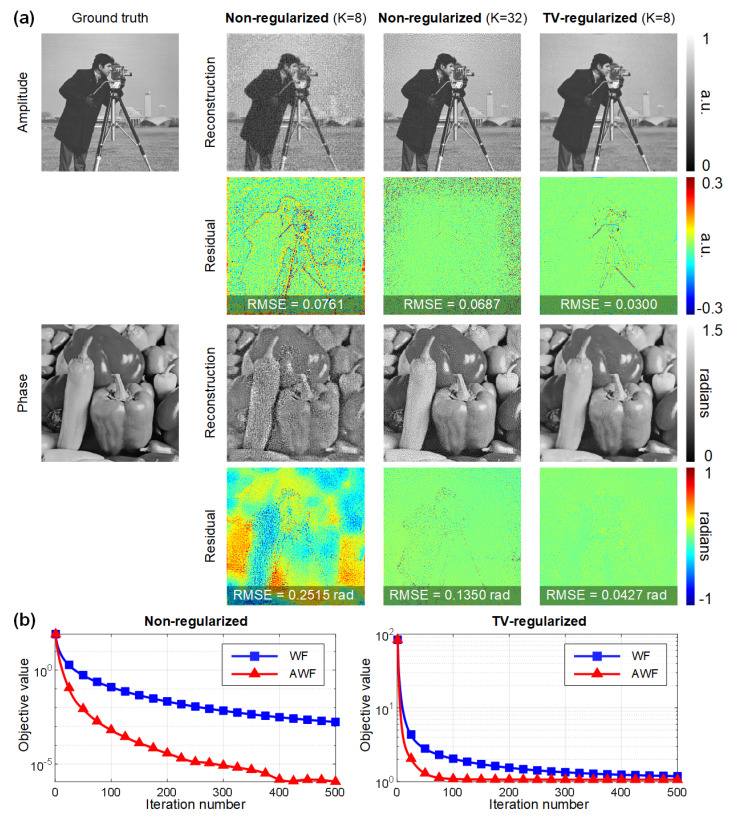
Simulation results. (**a**) Evaluation of the quality improvements by TV regularization. (**b**) Convergence curves of the AWF algorithm and the non-accelerated Wirtinger flow (WF) algorithm using K=8 diversity images.

**Figure 5 cells-11-01999-f005:**
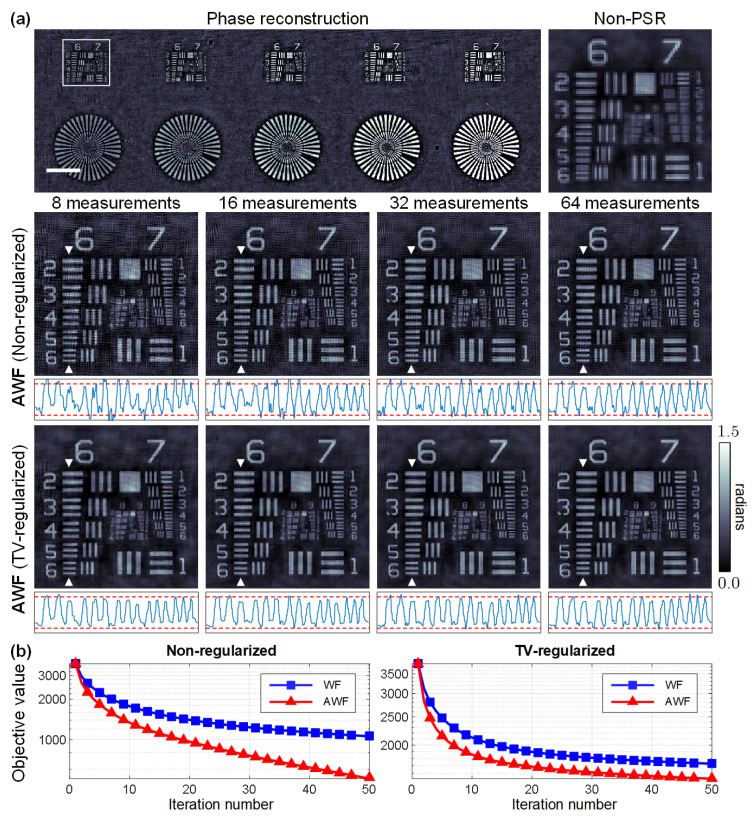
Experimental results. (**a**) Phase reconstruction of a quantitative phase target. The cross-sectional profiles are indicated by the triangular marks. The red doted lines indicate the ground truth phase induced by the structures. The scale bar is 200 μm. (**b**) Convergence curves of the algorithms using K=8 diversity images.

**Figure 6 cells-11-01999-f006:**
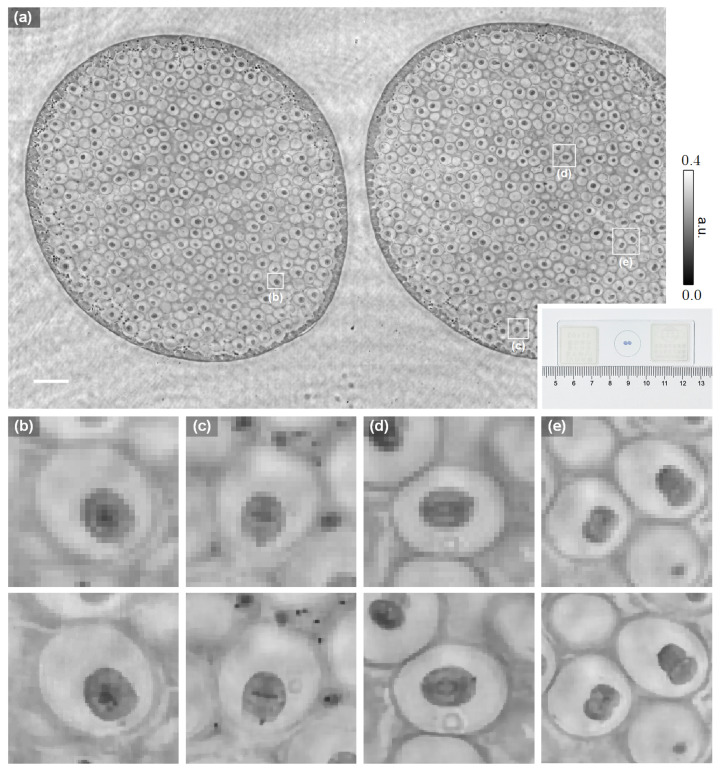
Experimental validation on biological samples.(**a**) Amplitude reconstruction of a section of the uterus of *parascaris equorum*. The inset shows an image of the stained tissue slide. (**b**–**e**) are the enlarged images of (**a**), corresponding to the prophase, metaphase, anaphase, and telophase of the mitosis, respectively. The upper and lower rows show the non-PSR and PSR reconstruction, respectively. The scale bar is 200 μm.

## Data Availability

The MATLAB code for AWF is available in [76]. Experimental data underlying the results presented in this paper may be obtained from the authors upon reasonable request.

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
