# Peer review of "Pixel Super-Resolution Phase Retrieval for Lensless On-Chip Microscopy via Accelerated Wirtinger Flow"

_cells, 2022, doi:10.3390/cells11131999_

Round 1
Reviewer 1 Report
The paper deals with the improvement of data reconstruction and retrieval in lensless digital holographic microscopy. In general, the material presented is interesting and worth to be published in Cells. However, before being eligible for publication the manuscript needs to address the following major concerns:
1. In the Introduction section, I feel a lack of more detailed discussion on other phase retrieval algorithms applied in inline digital holography. Just mentioning these techniques is insufficient. The comparison with other realizations of Wirtinger flow in other sections would also be beneficial.
2. In the current form, the paper is more appropriate for an optical journal. Since the paper is intended for the audience mostly with biological background the part describing the real biological experiment requires to be extended considerably.
In particular, the sketch of the experimental schematic would be informative, along with the data on experimental procedure, including sample preparation and experimental workflow. The idea on how long time it takes for the suggested algorithm to retrieve the final phase image using the specific computer would be helpful for users.
The detailed description of the results obtained is also necessary.
Specifically regarding Fig. 7.
- Why tissue staining was applied for phase imaging? And by what stain?
- Why the data is presented on reconstructed amplitude rather than phase?
- An explanation why the specific locations indicated by yellow arrows were chosen and what is shown by the yellows graphs, is necessary.
- Where are chromosomes in Fig. 7? Resolution of their shape is quite controversial.
- What is shown in the inset to Fig. 7,a?
- And last but not least, data quality in real experiment is quite lower than that using the model object shown in Fig. 6.
I have also some minor concerns that are as follows.
- Why phase modulation patterns in Fig. 3 are black and white instead of grayscale?
- In legends in Figs. 4 and 6 authors indicate “radius”. Should be “radians” I guess.
- There is also a misprint in the legend for Residuals in analysis of the Pepper image, and data interval is missing as well.
- Refs. 3 and 4 seem to cite the same paper with the erroneous data given in 3.
Reviewer 2 Report
The authors report about an approach for accelerated Wirtinger flow for pixel super-resolution phase retrieval. After an explanation of the underlying principles the method firstly is characterized by simulations and investigations on a resolution test target. Then the application is illustrated by experimental results on a biological specimen. In general, the manuscript is motivated, organized and include adequate references. The simulations and the experimental investigations appear to be accurately performed. The results are plausible and novel. However, the authors should consider revisions:
1. Title: Regarding the scope of the special issue and the journal “Cells” the authors should consider changing the manuscript title to a more application related one.
2. Abstract: In the abstract possible applications/application fields of the proposed method and their significance should be emphasized (see also comment 1).
3. Introduction and discussion / conclusions sections: The authors should describe and discuss their work more in context of possible application fields of quantitative phase imaging, e.g., as summarized in Nat. Photon. 12, 578-589 (2018), Sensors 13, 4170-4191 (2013), Neurophotonics 1, 020901 (2014), Histol. Histopathol. 33, 417-432 (2018) and references therein, and moreover should also discuss possible advances or limitations of their approach in comparison to other state-of-the art microscopy techniques with more details. This would significantly increase the impact of the manuscript.
4. Section “Experimental results”:
a. It should be explained and discussed why the data in Fig. 6b converge after lower iteration numbers than for the simulation results in Fig. 5. Moreover, the time consumption of the image generation (e.g., in seconds) should be specified and discussed briefly.
b. Fig. 6: Information about the utilized test target such (type, manufacturer, part number, etc.) should be provided.
c. Fig. 7: Additional information about the investigated sample should be provided, for example, the preparation/cutting protocol, section thickness, and if the sample has been stained during the preparation process. Moreover, the usage of the biological sample for the experiments should be motivated, for example, by a significant application.
d. The accuracy of phase retrieval should be discussed with details.
Round 2
Reviewer 1 Report
Authors addressed all my concerns and made adequate corrections to the manuscript. I recommend to publish the corrected version of the paper.